# Cross-View Contrastive Unification Guides Generative Pretraining for Molecular Property Prediction

### Junyu Lin
Shenzhen Institute for Advanced Study, University of Electronic Science And Technology of China
Shenzhen, China
linjunyuxx@gmail.com

### Yan Zheng
School of Computer Science and Engineering, University of Electronic Science and Technology of China
Chengdu, China
yan9zheng9@gmail.com

### Xinyue Chen
School of Computer Science and Engineering, University of Electronic Science and Technology of China
Chengdu, China
martinachen2580@gmail.com

### Yazhou Ren*
School of Computer Science and Engineering, University of Electronic Science and Technology of China
Chengdu, China
Shenzhen Institute for Advanced Study, University of Electronic Science And Technology of China
Shenzhen, China
yazhou.ren@uestc.edu.cn

### Xiaorong Pu
School of Computer Science and Engineering, University of Electronic Science and Technology of China
Chengdu, China
Shenzhen Institute for Advanced Study, University of Electronic Science And Technology of China
Shenzhen, China
puxiaor@uestc.edu.cn

### Jing He
Faculty of Medicine Biomedical Sciences, The University of Queensland
Brisbane, QLD, Australia
Jing.he@uq.edu.au

## Abstract

Multi-view based molecular properties prediction learning has received widely attention in recent years in terms of its potential for the downstream tasks in the field of drug discovery. However, the consistency of different molecular view representations and the full utilization of complementary information among them in existing multi-view molecular property prediction methods remain to be further explored. Furthermore, most current methods focus on generating global level representations at the graph level with information from different molecular views (e.g., 2D and 3D views) assuming that the information can be corresponded to each other. In fact it is not unusual that for example the conformation change or computational errors may lead to discrepancies between views. To addressing these issues, we propose a new **C**ross-**V**iew contrastive unification guides **G**enerative Molcular pre-trained model, call Mol-CVG. We first focus on common and private information extraction from 2D graph views and 3D geometric views of molecules, Minimizing the impact of noise in private information on subsequent strategies. To exploit both types of information in a more refined way, we propose a cross-view contrastive unification strategy to learn cross-view global information and guide the reconstruction of masked nodes, thus effectively optimizing global features and local descriptions. Extensive experiments on real-world molecular data sets demonstrate the effectiveness of our approach for molecular property prediction task.

## CCS Concepts

• **Computing methodologies → Learning latent representations**; • **Applied computing → Bioinformatics**.

## Keywords

Multi-View Learning, Molecular Property Prediction, Self-Supervised Learning

**ACM Reference Format:**
Junyu Lin, Yan Zheng, Xinyue Chen, Yazhou Ren, Xiaorong Pu, and Jing He. 2024. Cross-View Contrastive Unification Guides Generative Pretraining for Molecular Property Prediction. In *Proceedings of the 32nd ACM International Conference on Multimedia (MM '24), October 28-November 1, 2024, Melbourne, VIC, Australia.* ACM, New York, NY, USA, 9 pages. https://doi.org/10.1145/3664647.3681193

*Corresponding author.

## 1 Introduction

In the field of computer-aided drug discovery and development, the molecular property prediction (MPP) plays a vital role [3]. When applying deep learning models for MPP, effective learning and integration of different expression forms of molecules is essential for constructing accurate and comprehensive molecular characterizations [4, 11].

Recently, the field of multi-modal and multi-view learning has had a vast impact [25, 29], and we explore MPP from the perspective of multi-view learning. MPP can be categorized into two types of approaches, single-view and multi-view, according to the diversity of data perspectives. Single-view MPP focuses on a single type of data, such as fingerprints [10, 24], SMILES [6, 23], or 2D structure graphs [7, 8, 19, 34] to depict molecular properties. However, single-view based MPP methods are limited by a single data source and cannot

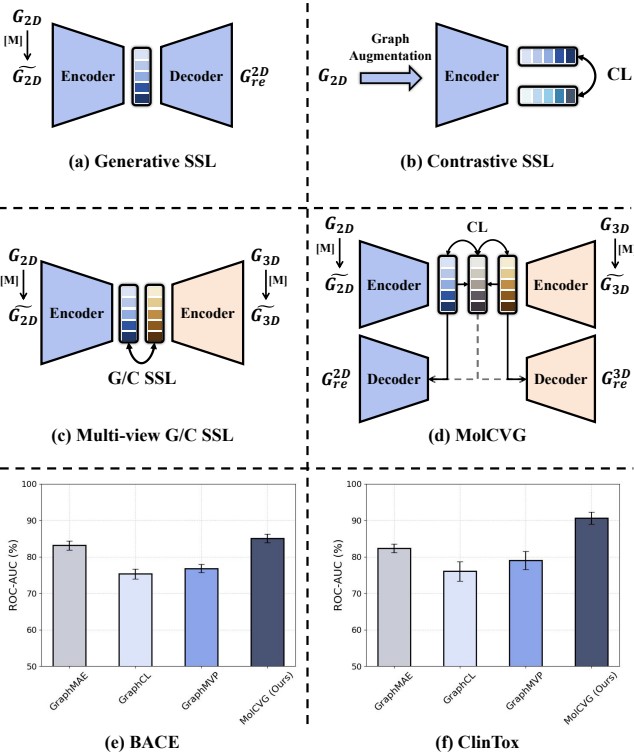

**Figure 1: Framework comparison of Single-view Generative SSL, Single-view Contrastive SSL, Multi-view Generative / Contrastive SSL (G/C SSL), and our MolCVG. [M] denotes the masking strategy, $\widetilde{G_{2D}}/\widetilde{G_{3D}}$ is obtained after $G_{2D}/G_{3D}$ is applied the masking strategy, and CL denotes Contrastive Learning. (e) and (f) are the performance comparisons of concrete implementations of the above four types of methods on BACE and ClinTox.**

to extract high-quality representations from a large amount of un-labeled molecular data, and then make them better serve the task of molecular property prediction through a fine-tuning process. In the pre-training phase, a crucial step is to explore an effective proxy task. Since multi-view data naturally contains both positive and negative pairs, contrastive self-supervised tasks have been widely adopted in previous GNN-based multi-view approaches for molecules [34, 37]. For example, Liu et al.[13] creates supervised signals for contrastive SSL on molecular 2D and 3D views.

However, such contrastive approaches are not always the op-timal choice. As shown in Figure 1, we summarize the prevailing self-supervised frameworks based on graph neural networks on molecular property prediction from single-view and multi-view perspectives. Figure 1(a) shows a typical graph mask autoencoder framework utilizing a single view for a rational design, and Graph-MAE is one of the representatives among this type of methods. Figure 1(b) is a typical graph contrastive learning framework, for example, GraphCL. Figure 1(c) is an approach based on combining contrastive and generative tasks under multi-view learning, and a specific representative method is GraphMVP. It is worth noting that the contrastive and generative tasks in Figure 1(b-c) are based on the alignment at the coarse-grained level under the molecular global domain or graph-level representation, which may be insuf-ficient for the learning of some discriminative information such as discriminative atoms or functional groups. Figure 1(e-f) demon-strates that the reconstruction of node features under single view achieves superior results to the contrastive methods. This confirms to some extent that there is potential for the learning of sufficient local details for performance improvement in molecular property prediction tasks.

Further, when exploring the limitations of multi-view molecular property prediction methods, there are problems shown below: (1) Ignoring view specificity and information filtering. Fewer studies have focused on the extraction of common information and view-specific private information for multiple views of a molecule at the same time, which may result in failing to fully utilize the comple-mentarity between views as well as effectively filtering out irrele-vant noise. Different views may contain their own unique biological or chemical signals that are critical for property-specific prediction, and simply merging views may overwhelm these signals, causing models to suffer from irrelevant or misleading information. (2) Over-reliance on global contrasts and reconstructions. Current methods generally focus on implementing contrast learning and mutual re-construction strategies on graph-level representations, emphasizing on global-scale feature alignment and consistent matching of cross-modal information [13, 34, 37]. such methods essentially perform similarity inference at a high level abstraction, assuming that the global information under different views can directly correspond to each other. This is not the case in practice. Local details may be neglected in such methods.

In this paper, we propose a cross-view contrastive unification guided generative pre-training method for molecular property pre-diction, called MolCVG, which utilizes cross-view learning to guide the graph autoencoder such generative pre-training. Specifically, the overview of MolCVG is shown in Figure 2, and we consider a 2D topological view and a 3D geometric view of the molecule. These two views provide information on the planar chemical structure

capture molecular properties in an all-encompassing way, which can easily lead to incomplete information and insufficient feature coverage. In comparison, multi-view MPP is aimed at effectively integrating information from multiple views through clever design and strategy to capture a broader range of contextual information [13, 15, 36]. Nevertheless, there may be noise, redundancy or irrele-vant information among the views, which may interfere with the learning process of common semantics when they are not properly addressed.

Therefore, exploring the further enhancement of multi-view MPP has received extensive attention. One of the main problems hindering MPP is the fact that obtaining labeled data are often challenging within the field of molecular biochemistry. Drawing on successful practices in the utilization of unlabeled data in other do-mains, especially extracting useful information by self-supervised learning techniques, provides insights into solving this dilemma [14, 21, 35, 38]. In this context, the framework of pre-training and fine-tuning has been applied in many studies. Combined with the fact that there are multiple view representations of molecules, op-timization of multi-view MPP is performed. This approach aims

of the molecule and the 3D spatial distribution of the molecule, respectively. MolCVG encodes these two views through an encoder after randomly masking the nodes. We then propose Common and Private Information Separation (CP-Info Separation) strategy to specifically learn and separate public and private information on molecular views. Weakening the interdependence between common and private information among different views to cope with the problem (1). Moreover, molecular representations are learned by MolCVG through the strategy like graph autoencoder. Unlike previous work, we propose cross-view contrastive unification to guide the reconstruction of the masked nodes to solve the problem (2). By deeply exploring and fusing the core common features of views, it not only contributes to a more accurate reconstruction of node features, but also effectively guards against the potential misdirection of view-specific information to the cross-view fusion process.

Our contributions in this work are as follows:

• We propose a new multi-view learning method for molecular property prediction called MolCVG. It deeply unites molecular 2d topological views and 3d geometric views to mine effective features.

• To the best of our knowledge, we are the first to propose molecular cross-view contrastive unification to guide node-level generative pretraining tasks. While ensuring that overall consistency is fully emphasized, we also focus on learning local detailed features.

• In MolCVG, we introduce the Common and Private Information Separation strategy to better capture molecular universality pattern and differences between molecular views.

• Extensive experiments demonstrate the effectiveness of MolCVG, achieving superior performance on multiple molecular property prediction benchmark datasets.

## 2 RELATED WORK

### 2.1 Single-View Molecular Property Prediction

Single-view molecular property prediction is mainly based on molecular representation in one dimension, SMILES strings, molecular fingerprints, 2D molecular structure graphs or 3D conformations, and other single data sources. In chemoinformatics, traditional chemical descriptors such as molecular fingerprints (e.g., ECFP, MACCS) [18, 28], etc., are utilized to establish quantitative relationships between molecular structures and properties by statistical learning methods (e.g., support vector machines, random forests, logistic regression, etc.). In recent years, deep learning techniques have played an important role in single-view molecular property prediction. High-level abstract features in molecules are learned through deep neural networks. Some sequence-based approaches with Transformer, etc. focus on the contextual information contained in the molecular sequences [23]. Some graph neural network-based methods directly deal with graphical structural representations of molecules. Hu et al. [8] designed a graph pretraining model that utilizes advanced graph neural network (GNN) techniques aimed to extract rich node and graph level representation information. You et al. [34] performed graph augmentation of molecular graphs to compare and learn from each other in a contrastive learning framework. Hou et al. [7] designed a generative

self-supervised method incorporating masked feature reconstruction and re-masking strategies. Liu et al. [12] introduced denoising to realize a pre-training framework for atom-to-distance denoising on molecular 3D views. Xia et al. [27] optimized the atomic vocabulary to mitigate the negative migration problem by alleviating the difference in the number of different atoms.

### 2.2 Multi-View Molecular Property Prediction

Multi-view molecular property prediction integrates multiple representations of molecules, including but not limited to 2D structures, 3D conformations, SMILES sequences, chemical descriptors, and possibly experimental data. It aims to reveal the complex relationship between molecular properties and structures more comprehensively by integrating information from multiple views. Chen et al. [2] combine Algebraic Graph Theory and Transformers to encode 3D molecular information into graph invariants and fuse the complementary molecular descriptors generated by the two to enhance the representation. Guo et al. [5] proposed a pre-training process by combining the SMILES representation of the molecule and IUPAC. Stärk et al. [22] optimize the matching degree between the learned 3D geometric structures and the 2D graphs. Liu et al. [13] design generative and contrastive interface tasks aimed at injecting information about 3D geometric views into 2D encoders. Zhu et al. [37] adopt four forms of molecular characterization to do comparative learning after fusion.

## 3 METHODOLOGY

### 3.1 Problem Setting

*3.1.1 Notation.* Given a molecular graph, we can denote it as $G = (A, V, C, X, E)$. Here $A \in \mathbb{R}^{N \times N}$ is the adjacency matrix of nodes with values taking only 0 and 1, $V = [v_1, v_2, \ldots, v_N]$ is the node set, $C = [c_1, c_2, \ldots, c_N]$ is the nodes' 3D coordinate matrix with each $c_i \in \mathbb{R}^{1 \times 3}$, $X \in \mathbb{R}^{N \times d}$ is denoted as a d-dimensional matrix of atom attributes, and $E \in \mathbb{R}^{N \times N \times d_E}$ records information about the properties of chemical bonds in the molecule where $d_E$ represents the dimension of the bond attribute. $N$ is the number of nodes.

*3.1.2 Problem Statement.* We aim to obtain effective molecular representations by training models through self-supervised learning. Considering the limitations of single-view representation of molecules, we employ a multi-view approach to enrich the molecular representation to contribute to more accurate molecular property prediction, which has been shown potential. Specifically, we consider 2D views and 3D geometric views of molecules, pre-trained on a large unlabeled dataset. Cross-view disparity contrasts in MolCVG drive multi-view fusion to guide the reconstruction of the masked nodes. For the fine-tuning phase, we finetune the pretrained 2D model on the downstream task dataset (2D molecular maps available). This method compensates for the limitations imposed by a single-view, while considering both local and global information.

### 3.2 Overview

In MolCVG, we intermingle two basic descriptions, 2D topological view and 3D geometric view, in a dual perspective within a

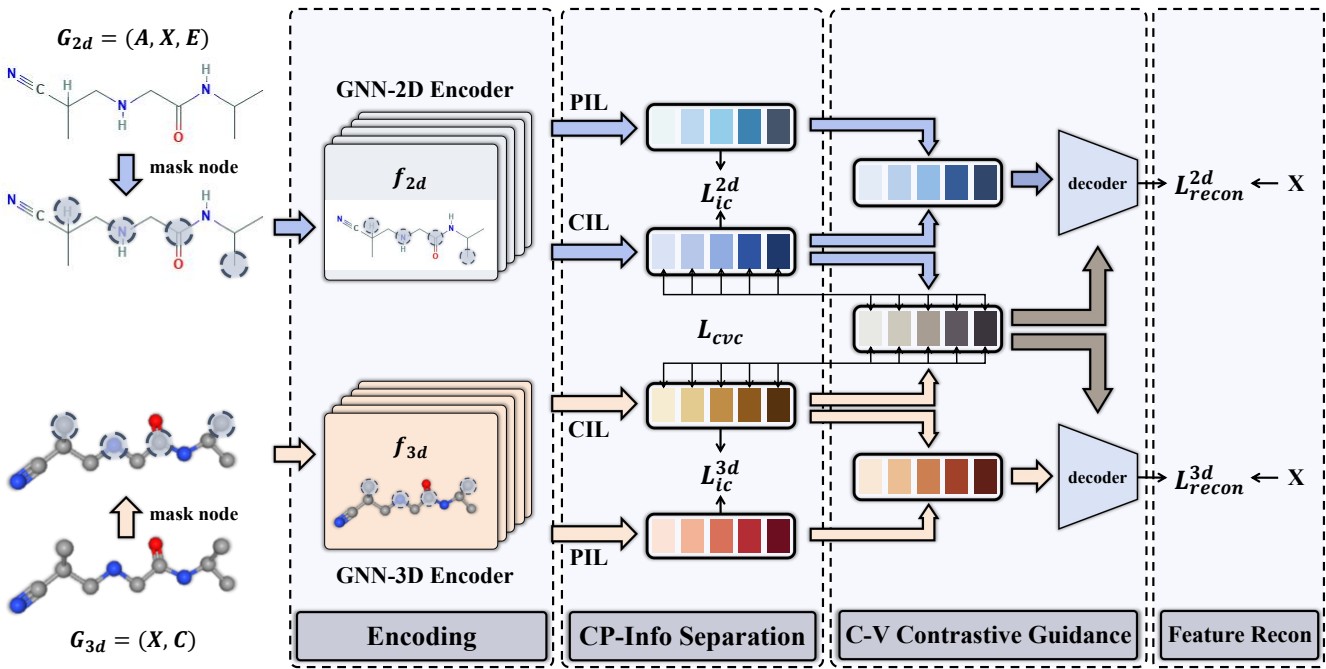

**Figure 2: General overview of MOlCVG. MolCVG encodes molecular 2D and 3D views, implements common and private Information Separation (Info Separation), and utilizes Cross-View Contrastive Unification Guidance (C-V Contrastive Guidance) for node feature reconstruction (Feature Recon).**

self-supervised learning framework. The framework of MolCVG is illustrated in Figure 2.

Inspired by GraphMAE [7], node feature reconstruction shows key influence in node and graph level classification tasks. Masking node features is achieved by randomly selecting a percentage (e.g., 0.25) of nodes in the graph data and masking their feature information. Operationally, we select a partial subset $\widetilde{V}$ within the node set $V$, and subsequently implement a hiding process for all the features of each node in the subset, with a predefined [MASK] marking symbol to accomplish this feature masking action. Its masked node attribute vector $\widetilde{x}_i$ can be obtained in the following formulation:

$$\widetilde{x}_i = \begin{cases} x_{[M]} & \text{if } v_i \in \widetilde{V}, \\ x_i & \text{otherwise.} \end{cases} \quad (1)$$

Masked node set $\widetilde{V}$ is determined by random sampling mechanism. For the node feature update mechanism, it can be clarified as follows: the process of feature reconstruction of any node $v_i$, is determined by the information of other nodes in the neighborhood to which the node belongs, in essence.

In 2D space, a molecular structure can be represented as a graph where atoms are considered as nodes and chemical bonds are depicted as edges connecting these nodes. We can consider the topological representation of the 2D molecular graph $G_{2D}$ as a mapping function $f_{2D} : G_{2D} \rightarrow \mathbb{R}^{N \times d_z}$, where $d_z$ is the dimension of the node latent representation. The function $f_{2D}$ receives three parameters, i.e., the adjacency matrix $A$, atom attributes vectors set $X$, and bond attributes vectors that may be integrated in the set $E$. For

a known 2D molecular graph, the representations $Z_{2D}$ are obtained by processing it through a 2D graph neural network (e.g. graph isomorphism network [30]):

$$Z_{2D} = f_{2D}(A, X, E). \quad (2)$$

In 3D space, molecular characterization involves describing the spatial arrangement of molecules, etc. Similarly, the mapping function $f_{3D} : G_{3D} \rightarrow \mathbb{R}^{N \times d_z}$ can be defined as a process of mapping from a 3D molecular graph $G_{3D}$ to a set of embedding vectors in a continuous vector space. Different from 2D, $f_{3D}$ has the ability to capture information such as the 3D spatial distances between atoms inside a molecule, which is crucial for understanding and predicting the physicochemical properties of molecules. $f_{3D}$ receives the atom attributes matrix $X$ and the atom 3D coordinates $C$ as parameters. For a given $G_{3D}$, its latent representation can be obtained by 3D graphical neural networks (e.g., Schnet [20]):

$$Z_{3D} = f_{3D}(X, C). \quad (3)$$

The pre-training objective of MolCVG is to decode the obtained latent representations of different molecular views to reconstruct the masked node features. However, due to the limitations of a single-view, we advocate utilizing data from multiple views to improve reconstruction performance. To address the situation where the information from different molecular views cannot actually correspond to each other, we attempt to learn the common and private information from multiple views. Specifically, effective discrimination and extraction of information is achieved by weakening the inter-correlation between common and private information.

Based on this foundation, we deeply contrast and fuse the common information of views from different sources, and these common information resources are utilized to guide the node feature reconstruction work. Such a strategy is effective because by deeply exploring and fusing the common core features shared by each view, node attributes can be reproduced more accurately while preventing private information from misleading cross-view fusion.

## 3.3 Common and Private Information Separation

Most molecular multi-view methods normally implement alignment operations between views only at the graph level (global) or at a coarse-grained scale. This alignment approach ignores the multi-level and complexity of the view's intrinsic structure and fails to fully reveal and utilize the fine-grained associations between views. Analyzing from the level of information extraction, existing methods tend to focus on mining the common information among views. Instead, they are not sufficient in mining the private information that is unique to a view and different from other views. To address the limitations faced by previous molecular multi-view approaches in extracting common information, we propose Common and Private Information Separation (CP-Info Separation) strategy to separate common and private information of different molecular views, as shown in CP-Info Separation in Figure 2. Also this facilitates the subsequent cross-view fusion strategy.

Specifically, we first obtain the latent representations (i.e., $Z_{2D}$ and $Z_{3D}$) by encoding on 2D and 3D views, respectively. Then, we map the latent representations to the corresponding common and private features by the Common Information Learning (CIL) layer and Private Information Learning (PIL) layer with independent parameter sets:

$$Z_{com}^i = \sigma(W_i^{com} Z_i + b_i^{com}), \quad i \in \{2D, 3D\}, \quad (4)$$

$$Z_{pri}^i = \sigma(W_i^{pri} Z_i + b_i^{pri}), \quad i \in \{2D, 3D\}, \quad (5)$$

where $Z_{com}^i$ represents common features, $Z_{pri}^i$ represents private features. $W^{com}$, $W^{pri}$, $b^{com}$, and $b^{pri}$ are the learnable parameters. $\sigma$ is an activation function.

For molecular 2D and 3D views of common and private information, the two parts of the information are intertwined to some extent. For example, the basic chemical structure in the common information and the energetics in the private information. Our goal is to make these two parts of information as unrelated as possible. Inspired by some previous work [1, 9, 16], the independence assumption is introduced to force the common information to be statistically independent or approximately independent from the private information in the feature space. The inter-correlation loss is set to weaken the statistical correlation between the common and private information by minimizing the Pearson's correlation coefficient of them:

$$L_{ic} = exp\left(\frac{1}{2}\left[\frac{|Cov(Z_{com}^{2D}, Z_{pri}^{2D})|}{\sigma_{Z_{com}^{2D}} \sigma_{Z_{pri}^{2D}}} + \frac{|Cov(Z_{com}^{3D}, Z_{pri}^{3D})|}{\sigma_{Z_{com}^{3D}} \sigma_{Z_{pri}^{3D}}}\right]\right), \quad (6)$$

where $Cov(\cdot)$ denotes covariance and $\sigma$ denotes standard deviation. Non-linear amplification of the inter-correlation coefficient

is performed by the exponential function, which exerts stronger penalties on higher inter-correlation strengths.

## 3.4 Cross-view contrastive unification

Previous approaches have built on the premise that there are clear correspondences between views, but intuitive contrast learning or mindless fusion of views without rigorous filtering tends to confuse perceptions of inherent consistency within the views. Therefore, we propose a cross-view contrastive unification fusion strategy that aims to accurately extract and fuse common information from both 2D and 3D views. This establishes the essential foundation for guiding node feature reconstruction.

We first concatenate the common information vectors $Z_{com}^{2D}$ and $Z_{com}^{3D}$ of the two views to construct the joint information vector $J_C = Z_{com}^{2D} || Z_{com}^{3D}$, where $||$ is the concatenation operation. $J_C$ is processed by a multilayer perceptron (MLP) to obtain the cross-view fusion representation $Z_{fuse}$. It fuses the multi-dimensional information of $Z_{com}^{2D}$ and $Z_{com}^{3D}$, and realizes the knowledge cross-fertilization between the views through nonlinear mapping:

$$Z_{fuse} = M_\eta(Z_{com}^{2D} || Z_{com}^{3D}), \quad (7)$$

where $M_\eta$ is a learnable matrix.

We introduce contrastive learning [17] to discover and emphasize those elements that are consistent. First, we apply the readout function to $Z_{fuse}$, $Z_{com}^{2D}$, $Z_{com}^{3D}$ to obtain $H^{fuse}$, $H^{2D}$, $H^{3D}$ respectively. Taking the 2D view as an example, we mark the positive pairs as $\{(h_i^{fuse}, h_{com,j}^{2D}) | i = j\}$, and negative pairs as $\{(h_i^{fuse}, h_{com,j}^{2D}) | i \neq j\}$. For the contrastive loss under the 2D view, the calculation formula can be expressed as follows:

$$\phi(h_i^{fuse}, h_{com,i}^{2D}) = \frac{h_i^{fuse} \cdot h_{com,i}^{2D}}{||h_i^{fuse}|| \cdot ||h_{com,i}^{2D}||}, \quad (8)$$

$$L_{2D} = -\frac{1}{N}\sum_{i=1}^N \log\left(\frac{exp(\phi(h_i^{fuse}, h_{com,i}^{2D})/\tau)}{\sum_{j=1}^N exp(\phi(h_i^{fuse}, h_{com,j}^{2D})/\tau)}\right), \quad (9)$$

where $\phi(\cdot)$ denotes the dot product similarity between the two vectors and $\tau$ is the temperature hyperparameter. Similarly, we calculate the contrastive loss for 3D views:

$$L_{3D} = -\frac{1}{N}\sum_{i=1}^N \log\left(\frac{exp(\phi(h_i^{fuse}, h_{com,i}^{3D})/\tau)}{\sum_{j=1}^N exp(\phi(h_i^{fuse}, h_{com,j}^{3D})/\tau)}\right). \quad (10)$$

The total cross-view contrastive unification loss function is as follows:

$$L_{cvc} = \frac{1}{2}(L_{2D} + L_{3D}). \quad (11)$$

## 3.5 Training Objective

The pre-training goal of MolCVG is to reconstruct mask node features. From GraphMAE, we recognize the potential of this generative pre-training and the existence of some improvement possibilities. It is limited by a single data source to fully capture molecular features, which easily leads to incomplete information and insufficient feature coverage. For this reason, we propose Cross-view contrastive unification to guide node feature reconstruction while

focusing on global consistency and local details learning. We first concatenate the common and private representations, and then obtain the pre-reconstructed node representations $Z_{pr}^{\delta}$ through an Linear funtion:

$$Z_{pr}^{\delta} = W^{\delta}(Z_{com}^{\delta}||Z_{pri}^{\delta}) + b^{\delta}, \quad \delta \in \{2D, 3D\}, \tag{12}$$

where $W^{\delta}$ and $b^{\delta}$ are the learnable weights and biases, respectively. This is currently still in the single-view case. With multi-view enhancement, the features of cross-view contrastive unification are leveraged to enrich the recognition capability of the model. Specifically, the knowledge or information provided by the cross-view fusion feature $Z_{fuse}$ guides the pre-reconstructed node feature set $Z_{pr}^{\delta}$ for reconstruction adaptation:

$$Z_g^{\delta} = M_{\theta}^{\delta}(Z_{pr}^{\delta}||Z_{fuse}), \quad \delta \in \{2D, 3D\}, \tag{13}$$

where $M_{\theta}^{\delta}$ is a single-layer neural network. Finally, we decode the cross-view guided reconstruction features $P$ to obtain the re-constructed outputs. Formally, we denote the reconstructed node feature set as $Z_{re}^{\delta} = f_d(Z_g^{\delta})$. $f_d$ is implemented as a single-layer MLP. The node reconstruction loss is calculated by computing the cosine similarity between the reconstructed node feature set $Z_{re}^{\delta}$ and the corresponding original node features. It is worth noting that only mask nodes are computed for node reconstruction loss. The loss can be expressed as follows:

$$L_{recon} = \frac{1}{|\Omega|} \frac{1}{|\widetilde{V}|} \sum_{\delta \in \Omega} \sum_{v_i \in \widetilde{V}} (1 - \frac{x_i^T Z_{re}^{\delta}}{||x_i|| \cdot ||Z_{re}^{\delta}||}), \tag{14}$$

where $\Omega = \{2D, 3D\}$, $|\Omega|$ represents the total number of elements in the set $\Omega$, $|\widetilde{V}|$ denotes the total number of elements contained in the mask node set $\widetilde{V}$.

In the pre-training phase the overall loss of MolCVG consists of three components: the node reconstruction loss $L_{recon}$, the inter-correlation loss $L_{ic}$ and the cross-view contrastive unification loss $L_{cvc}$. The formula is as follows:

$$L = L_{recon} + \alpha L_{cvc} + \beta L_{ic}, \tag{15}$$

where $\alpha$ and $\beta$ are hyperparameters for regulating weights.

The fine-tuning phase, the pretrained 2D GNN model is applied for fine-tuning on specific downstream tasks. At this point, it relies only on the 2D molecular map data. MolCVG is fine-tuned on each of the eight benchmark molecular datasets used for the classification task. The probability of the output molecule $i$ to be determined as a positive class is denoted as $p_i$ after the encoded features are processed by the prediction header. The loss of the prediction and the target label is calculated as follows:

$$L_{ft}(p, y) = -\frac{1}{N} \sum_{i=1}^{N} \left[ y_i \log(\sigma(p_i)) + (1 - y_i) \log(1 - \sigma(p_i)) \right], \tag{16}$$

where $y_i$ is the label of the molecule $i$. MolCVG implemented optimizations for 2D characterization of molecules in a generative pre-training task, and subsequent ablation studies demonstrated these enhancements.

**Table 1: Summary of the benchmark datasets.**

| Category | Dataset | # Molecule | # Tasks |
|---|---|---|---|
| Physiology | BBBP | 2,039 | 1 |
| Physiology | ClinTox | 1,478 | 2 |
| Physiology | Tox21 | 7,831 | 12 |
| Physiology | SIDER | 1,427 | 27 |
| Physiology | ToxCast | 8,575 | 617 |
| Biophysics | BACE | 1,513 | 1 |
| Biophysics | HIV | 41,127 | 1 |
| Biophysics | MUV | 93,087 | 17 |

## 4 EXPERIMENT

### 4.1 Dataset

The pre-training dataset we used was obtained from the PubChemQC database, which contains about 4 million molecules with 3D geo-metric configurations. These data were accurately computed by using Density Functional Theory (DFT). These data are optimized by Molecule3D [32] to be a suitable form for deep learning model processing. For pre-training, we selected a subset containing 1 mil-lion molecular 3D geometric conformation.

For the fine-tuning dataset, we evaluated the performance of MolCVG on 8 public datasets on MoleculeNet [26], including BACE, BBBP, ClinTox, Tox21, SIDER, ToxCast, MUV, and HIV, which are widely adopted in molecular and drug representation learning and property prediction tasks. For each benchmark dataset, the scaffold splitting scheme is applied to divide them. These molecules are assigned to the training set, validation set, and test set in the ratio of 80%, 10%, and 10%. The statistics for these eight datasets are summarized in Table 1.

### 4.2 Experimental Setup

**Baselines**. To demonstrate the effectiveness of MolCVG, we selected eight baselines as benchmark references. Among them are ContextPred [8], AttrMasking [8], GraphCL [34], JOAO [33], GraphLoG [31], GraphMAE [7] and Mol-BERT [27] for single-view molecular property prediction, and GraphMVP [13], 3D Infomax [22], MEMO [37] for multi-view molecular property prediction.

**Evaluation metric**. During the evaluation process, we follow previous work to adopt the area under the receiver operating char-acteristic curve (ROC-AUC) as the primary metric. For each experi-ment, we ensure that three independent executions were performed using different seeds and finally recorded the mean of each experi-ment and its corresponding standard deviation as the presentation of the experimental results.

**Implementation details**. In the process of constructing molecu-lar 2D view representations, we draw on the successful practices of previous studies and choose the Graph Isomorphism Network [30] with a five-layer structure as the core to construct the backbone model. For encoding the molecular 3D conformations, the Schnet [20] model is adopted as the basic backbone to effectively encode and analyze the molecular 3D views. In the pre-training phase, we set the masking ratio to 0.25. The Adam optimizer is adopted for the pre-training process of 100 training epoch, initializing the learning rate to $8 \times 10^{-4}$ and setting the batch size to 256. We set

**Table 2: For the eight molecular property prediction tasks on the MoleculeNet benchmark, the best and second best ROC-AUC (%) results are highlighted in bold and underlined, respectively.**

| Method | BACE | BBBP | SIDER | Tox21 | ClinTox | ToxCast | MUV | HIV | Avg |
|---|---|---|---|---|---|---|---|---|---|
| ContextPred [8] | 79.6(1.2) | 64.3(2.8) | 60.9(0.6) | 75.7(0.7) | 65.9(3.8) | 63.9(0.6) | 75.8(1.7) | 77.3(1.0) | 70.4 |
| AttrMasking [8] | 79.3(1.6) | 64.3(2.8) | 61.0(0.7) | 76.7(0.4) | 71.8(4.1) | 64.2(0.5) | 74.7(1.4) | 77.2(1.1) | 71.1 |
| GrapgCL [34] | 75.3(1.4) | 69.7(0.7) | 60.5(0.9) | 73.9(0.7) | 76.0(2.7) | 62.4(0.6) | 69.8(2.7) | 78.5(1.2) | 70.8 |
| JOAO [33] | 77.3(0.5) | 70.2(1.0) | 60.0(0.8) | 75.0(0.3) | 81.3(2.5) | 62.9(0.5) | 71.7(1.4) | 76.7(1.2) | 71.9 |
| GraphLoG [31] | 83.5(1.2) | 72.5(0.8) | 61.2(1.1) | 75.7(0.5) | 61.2(1.1) | 63.5(0.7) | 76.0(1.1) | 77.8(0.8) | 73.4 |
| GraphMAE [7] | 83.1(0.9) | 72.0(0.6) | 60.3(1.1) | 75.5(0.6) | 82.3(1.2) | 64.1(0.3) | 76.3(2.4) | 77.2(1.0) | 73.8 |
| GraphMVP [13] | 76.8(1.1) | 68.5(0.2) | 62.3(1.6) | 74.5(0.4) | 79.0(2.5) | 62.7(0.1) | 75.0(1.4) | 74.8(1.4) | 71.7 |
| 3D InfoMax [22] | 79.4(1.9) | 69.1(1.0) | 60.6(0.7) | 74.5(0.7) | 79.9(3.4) | 64.4(0.8) | 76.2(1.4) | 76.1(1.3) | 72.5 |
| MEMO [37] | 82.6(0.3) | 71.6(1.0) | 61.2(0.6) | 76.7(0.4) | 81.6(3.7) | **64.9(0.8)** | 78.5(0.5) | 78.3(0.4) | 74.4 |
| Mole-BERT [27] | 80.8(1.4) | 71.9(1.6) | 62.8(1.1) | **76.8(0.5)** | 78.9(3.0) | 64.3(0.2) | 78.6(1.8) | 78.2(0.8) | 74.0 |
| MolCVG (our) | **85.0(1.2)** | **72.9(1.2)** | 62.9(1.3) | 76.7(0.7) | **90.6(1.7)** | 64.4(0.8) | **78.9(1.0)** | **79.2(0.8)** | 76.3 |

**Table 3: Ablation results of MolCVG loss function.**

| MolCVG Loss | | | dataset | | |
|---|---|---|---|---|---|
| $L_{recon}$ | $L_{cvc}$ | $L_{ic}$ | BACE | BBBP | ClinTox |
| √ | - | - | 83.1(0.9) | 72.0(0.6) | 82.3(1.2) |
| √ | - | √ | 83.5(1.0) | 71.1(0.6) | 86.5(1.2) |
| √ | √ | - | 83.7(0.7) | 72.3(0.8) | 86.5(1.8) |
| √ | √ | √ | 85.0(1.2) | 72.9(1.2) | 90.6(1.7) |

**Table 4: Ablation results of MolCVG's strategies**

| strategies | | dataset | | |
|---|---|---|---|---|
| C-V Contrastive Guidance | CP-Info Separation | BACE | BBBP | ClinTox |
| - | - | 83.1(0.9) | 72.0(0.6) | 82.3(1.2) |
| - | √ | 83.2(0.9) | 71.8(0.6) | 86.1(1.6) |
| √ | - | 84.3(0.7) | 72.4(0.9) | 86.6(2.1) |
| √ | √ | 85.0(1.2) | 72.9(1.2) | 90.6(1.7) |

the mask ratio and temperature coefficient by default to 0.25 and 0.1 respectively. In addition, we apply a cosine annealing learning rate scheduling strategy to dynamically adjust the learning rate throughout the pre-training process. In the fine-tuning phase, the Adam optimizer is still followed for model optimization. We determine the learning rate, dropout, batch size, and epoch with grid search. All experiments are done in RTX 3080 GPU environment.

## 4.3   Results and Analysis

Table 2 summarizes the performance of the comparison methods for molecular property prediction on the eight datasets. It is demonstrated that our MolCVG model exhibits excellent performance on all 8 downstream task datasets, with 6 of them achieving the best results and realizing an absolute performance improvement of up to 1.9% in average performance compared to the existing methods. The outstanding results validate the superiority of our proposed method. It shows that MolCVG can better capture the commonalities and differences between molecular 2D and 3D views, and it can fully utilize the multi-view information of molecules.

Compared to the baseline of single-view molecular property prediction, MolCVG outperforms almost all baselines. Specifically, for example, when evaluated on the BACE and ClinTox datasets, MolCVG performed significantly better by 1.5% and 8.3%, respectively. We found that MolCVG slightly underperforms Mol-Bert on the Tox21 dataset by 0.1%, which may be due to dataset-specificity and randomization factors, but both also have comparable performance. Overall an absolute improvement of 2.3% was obtained on average. Single-view molecular property prediction method emphasizes the deep mining and learning of a single-view and it fails to enjoy

the complementary information from multi-views. GraphMAE and GraphCL are the typical generative and contrastive approaches, respectively, which both produce suboptimal results compared to MolCVG. This goes some way to show that it is quite feasible for our cross-view contrastive unification fusion to guide generative tasks. These results demonstrate that MolCVG has superior hidden representation learning capability and can effectively handle the coordination of multi-view information under the molecular property prediction task.

Compared with the multi-view molecular property prediction baseline, MolCVG also outperforms almost all multi-view MPP baselines. Specifically, an overall average improvement of 1.9% is obtained. We notice that MolCVG underperforms MEMO by 0.5% on the Toxcast dataset. We speculate that this is a result of MEMO utilizing more complementary information from the four molecular views for processing, improving its performance on the Toxcast dataset. In addition, GraphMVP and 3D Infomax both utilize 2D and 3D views for self-supervised methods. The experimental results show that MolCVG outperforms them both. It resonates with our motivation that there is a performance bottleneck in global contrast and reconstruction based on the assumption that multi-view information can be matched one-to-one, which may overlook important local details.

## 4.4   Ablation Study

*4.4.1   The effectiveness of the MolCVG's loss function.* In this subsection, we investigate the node reconstruction loss $L_{recon}$, the

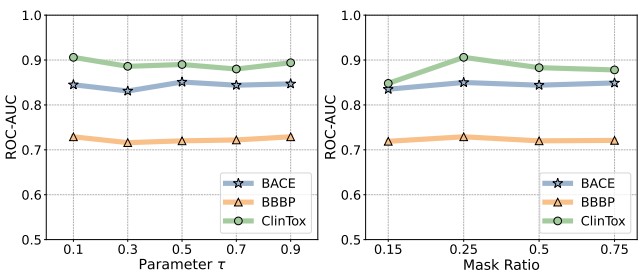

Figure 3: Ablation experiments of temperature hyperparameter $\tau$ and mask ratio on datasets BACE, BBBP, and ClinTox.

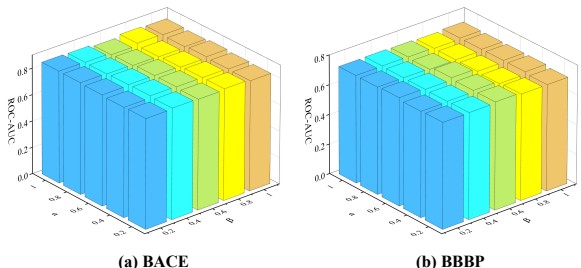

(a) BACE          (b) BBBP

Figure 4: Ablation study of hyperparameters $\alpha$ and $\beta$ used to regulate weights in the loss function.

inter-correlation loss $L_{ic}$ and the cross-view contrastive unification loss $L_{cvc}$ . We report the performance of each loss component on the three datasets BACE, BBBP, and ClinTox in Table 3. The experimental results show that our method with a complete loss function achieves optimal performance. Specifically, it improves 1.9%, 0.9%, and 8.3% on the BACE, BBBP, and ClinTox datasets, respectively, compared to only with node reconstruction loss. This indicates that the loss of our method is significantly effective. Further, we observe a performance decrease on the BBBP dataset when our method lacks the cross-view contrastive unification fusion loss, although the overall performance is improved. This corresponds to our previous observational analysis, where direct fusion without differentiating the data across views may misleading information cause interference. In addition, the complete MolCVG obtains a larger increase in performance compared to the case where inter-correlation loss is lacking. This also suggests that neglecting view specificity and information filtering limits model to learn effective representations.

*4.4.2 The effectiveness of the strategies of MolCVG.* Table 4 demonstrates the effectiveness study of two strategies for MolCVG. The experimental results show that our proposed method achieves superior performance. Note that there are differences in the setup of this strategy ablation and loss ablation, and we aim to explore the effects of these two strategies. Specifically, the C-V Contrastive Guidance setting item in Table 4 refers to whether the fused features obtained through cross-view contrastive unification fusion are used to guide node feature reconstruction, but the cross-view contrastive unification loss function remains. CP-Info Separation refers to whether to extract and separate the common and private information of each view or not, unlike loss ablation this setting eliminates CIL and PIL, and at the same time inter-correlation loss also its cannot be constructed. This is because it is not possible to preserve the inter-correlation loss function without adopting an information separation strategy. We can observe that both strategies contribute to the performance of the model. It is worth noting that only lacking the CP-Info Separation strategy or the C-V Contrastive Guidance, although the performance is improved compared to the version that lacks both, there is still a large gap compared to the full version of MolCVG. In particular, on the ClinTox dataset, there is still a 4% significant gap. This phenomenon demonstrates that MolCVG combines the two strategies well and achieves significant results. It also shows that the successful integration of these two strategies

in the MolCVG framework effectively addresses and solves the two challenges mentioned in above.

*4.4.3 Effect of Mask Ratio.* The plot on the right of Figure 3, shows the effect of the mask ratio. In most cases, when a low mask ratio is adopted for the feature reconstruction task, it is not challenging enough to motivate the model to learn beneficial features. If the mask ratio is set too high, it may result in certain key features not being effectively recovered, which may cause performance degradation. As the mask ratio increases from 0.15 to 0.25, the model performance shows significant improvement, demonstrating that moderately increasing the difficulty of feature reconstruction contributes to the model's ability to learn richer and more valuable intrinsic patterns. However, as the mask ratio continues to increase to 0.75, there is a slight decrease in the overall performance at first both, after that the model shows a relatively stable trend. Based on experience we set the mask ratio to 0.25.

*4.4.4 Parameter sensitivity analysis.* We performed an ablation study of the hyperparameters in MolCVG, and the detailed results are shown in Figures 3 and 4. In Eq. (15), the trade-off coefficient hyperparameters $\alpha$ and $\beta$ are included. We set $\alpha, \beta \in [0.2, 1.0]$ with an interval of 0.2, respectively, to explore the effects of these two parameters in MolCVG. As shown in Figure 4, our experimental results demonstrate that our method is robust to the choice of $\alpha$ and $\beta$. Moreover, we perform ablation experiments on the temperature coefficient $\tau$ in Eq. (9) and Eq. (10). We set $\tau \in [0.1, 0.9]$ with an interval of 0.2 to explore the effect of temperature hyperparameter in MolCVG. As shown in the left plot of Figure. 3, there is good robustness of MolCVG to $\tau$ over the experimental range we set.

## 5 CONCLUSION

In this paper, we present MolCVG, a cross-view contrastive unification guides generative pre-training method , which considers molecular 2d topological views and 3d geometrical views jointly for molecular property prediction. In MolCVG, after encoding the 2D and 3D views of molecule, the common and private information are extracted and separated from the different views of molecule. Focusing on global consistency while taking into account the preservation and learning of local details, MolCVG designs a cross-view contrastive unification to guide node feature reconstruction. Extensive experiments on eight public molecular property prediction benchmarks demonstrate the superiority of our method.

# Acknowledgments

This work was supported in part by Sichuan Science and Technology Program (No. 2024NSFSC1473) and in part by Shenzhen Science and Technology Program (Nos. JCYJ20230807115959041 and JCYJ20230807120010021).

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
