# OpenReview forum: "Cross-view Contrastive Unification Guides Generative Pretraining for Molecular Property Prediction"
_acmmm.org/ACMMM/2024/Conference — MM2024 Poster_

### Official Review · Reviewer_iRKn · 2024-05-19

**Rating:** 4
**Confidence:** 3

**Summary:**

This work proposed MolCVG, a method for molecular property prediction that leverages cross-view contrastive unification and generative pretraining. The method integrates 2D topological and 3D geometric views of molecules to enhance the prediction accuracy of molecular properties. The authors demonstrate that MolCVG outperforms existing methods on multiple benchmark datasets, showing the effectiveness of their approach.

**Strengths:**

1.  The paper is of high quality and is clearly written.

2.  The key observations and basic ideas behind this method are particularly noteworthy. By separating common and private information from different molecular views, MolCVG effectively addresses the issue of noise and inconsistency between views. The cross-view contrastive unification strategy ensures that the decoder benefits from the rich, discriminative features learned during the contrastive phase.

3.  The experimental results are comprehensive, and confirm the merits of the proposed method.

**Limitations:**

1.  The improvements are mainly observed on the ClinTox and BACE datasets. On other datasets, the performance is only marginally better or sometimes slightly worse compared to existing methods.

2.  The pre-training dataset used in MolCVG contains 1 million molecular 3D geometric conformations, which is significantly larger than the datasets used in previous works such as GraphMVP and MEMO. While the large dataset is a strength, it also raises questions about the generalizability and scalability of the approach. The manuscript does not sufficiently discuss how MolCVG performs with smaller datasets or the computational resources required for handling such a large dataset. A comparison of performance on smaller pre-training datasets or a discussion on the impact of dataset size on model performance would provide a more comprehensive evaluation.

**Suitability:**

2

---

### Official Review · Reviewer_aRJF · 2024-05-23

**Rating:** 4
**Confidence:** 2

**Summary:**

## Summary
The paper presents a novel method, MolCVG, which combines multi-view learning approaches to improve molecular property prediction. The core innovation lies in integrating 2D topological and 3D geometric views of molecules through a process involving encoding, information separation, and cross-view contrastive guidance. The methodology encompasses several components:

1. **Encoding of molecular views**: Utilizing Graph Neural Networks (GNNs) to encode both 2D and 3D views into feature representations.
2. **Common and Private Information Separation**: Distinct layers are used to extract common (universal across views) and private (unique to each view) information, enhancing the adaptability of the model to various molecular structures.
3. **Cross-View Contrastive Unification**: Employing cross-view learning to enhance feature reconstruction and ensure that information learned from one view can enhance the learning from another.

### Questions:


1. **Statistical Independence in Feature Separation**:
How do you quantitatively measure and ensure the statistical independence between common and private features? What specific metrics or tests are used to evaluate the effectiveness of this separation in practical scenarios?

2. **Model Complexity and Efficiency**:
Given the apparent complexity of the MolCVG model with its multi-component structure involving encoders, decoders, and separate pathways for common and private information, what are the computational requirements? How does the model's performance scale with increasing dataset sizes or molecular complexity?

3. **Generalisation and Overfitting**:
What measures are in place within MolCVG to prevent overfitting, especially given the deep and complex architecture? Are there specific regularisation techniques or model validation strategies that have proven effective?

4. **Fine-Tuning and Adaptability**:
During the fine-tuning phase, how adaptable is the model to different types of downstream tasks? Are there specific examples where MolCVG has shown significant improvements over existing models?

**Strengths:**

1. **Innovative Integration of Multi-View Data**: The methodology effectively utilizes the complementary nature of 2D and 3D molecular data, which is a significant advancement in molecular modeling. This integration helps in capturing both the structural and relational aspects of molecules more comprehensively than using either view in isolation.

2. **Robust Information Separation Strategy**: The approach of separating common and private information is methodologically sound and innovative. This separation allows the model to better generalize across different molecular datasets by leveraging universal features while preserving view-specific details that might be crucial for certain properties.

3. **Enhanced Learning Through Cross-View Contrastive Guidance**: This component is particularly notable for its potential to refine feature learning by using insights gained from one view to inform the feature reconstruction in another. This could potentially lead to more accurate predictions by ensuring that the model fully exploits all available information.

**Limitations:**

1. **Conceptual Overlap with Existing Techniques**:
While MolCVG’s approach is framed as innovative, parts of its methodology closely resemble existing strategies in machine learning that involve feature integration and contrastive learning, potentially diluting its perceived novelty.

2. **Complexity and Computational Demands**: The complex structure, while powerful, could pose challenges in terms of computational efficiency and scalability, especially when deployed in resource-limited settings.

3. **Empirical Validation Needs**: While the methodological innovations are clear, the actual performance benefits of these complex strategies would benefit from more extensive empirical validation across diverse and large-scale datasets.

4. **Handling of Conflicting Information**: The methodology does not explicitly address how contradictions between 2D and 3D information are resolved during the feature reconstruction phase.

**Suitability:**

2

---

### Official Review · Reviewer_dXW4 · 2024-05-24

**Rating:** 3
**Confidence:** 3

**Summary:**

This paper presents MolCVG, a novel method for molecular property prediction that leverages cross-view contrastive learning and generative pretraining. The approach utilizes two different views of molecular data: the 2D topological view and the 3D geometric view. MolCVG aims to address limitations in current multi-view molecular property prediction methods, specifically the issues of view specificity and the loss of local detail information. Authors introduced a cross-view contrastive unification to guide node-level generative pretraining, ensuring both global consistency and local detail preservation. Extensive experiments demonstrate that MolCVG achieves state-of-the-art performance on multiple molecular property prediction benchmark datasets, surpassing existing methods in most cases.

**Strengths:**

1. The paper proposes a novel method that integrates cross-view contrastive learning with generative pretraining.
2. The CP-Info Separation strategy is a strong point as it effectively distinguishes between common and unique information from different views, enhancing the model’s ability to utilize complementary information from both 2D and 3D representations.

**Limitations:**

1. The proposed model’s complexity and computational requirements are not discussed in depth. Given that it integrates multiple views and involves sophisticated strategies like CP-Info Separation, it is likely to be computationally intensive.
2. While the paper demonstrates effectiveness on benchmark datasets, there is limited discussion on the scalability of the approach to larger datasets or in real-world applications. The dataset scales shows in Table 1 are small.
3. The performance slightly varies across different datasets.  (e.g., underperforming on the Tox21 dataset compared to Mol-Bert), suggesting that the method may have dataset-specific dependencies​.
4. The contrastive learning strategy is common in biological area, such GearNet[1].
5. As for the hyperparamerts, 𝛼 and 𝛽, and the the temperature coefficient $\tau$, why the model is robust to the choices of these parameters?

[1] Zhang, Z., Xu, M., Lozano, A. C., Chenthamarakshan, V., Das, P., & Tang, J. (2024). Pre-training protein encoder via siamese sequence-structure diffusion trajectory prediction. Advances in Neural Information Processing Systems, 36.

**Suitability:**

1

---

### Meta-Review · Area_Chair_ZmUM · 2024-06-29

**Recommendation:** Accept (Poster)
**Confidence:** 3

**Metareview:**

This paper received ba, ba, br from all of reviewers after rebuttal. I have read the paper and believe the paper is of high quality. I am happy to recommend to accept this paper. Please carefully revise the final manuscript according to the comments and discussions.